# Finasteride-Induced Inhibition of 5α-Reductase Type 2 Could Lead to Kidney Damage—Animal, Experimental Study

**DOI:** 10.3390/ijerph16101726

**Published:** 2019-05-16

**Authors:** Mirza Saim Baig, Agnieszka Kolasa-Wołosiuk, Anna Pilutin, Krzysztof Safranow, Irena Baranowska-Bosiacka, Joanna Kabat-Koperska, Barbara Wiszniewska

**Affiliations:** 1Department of Histology and Embryology, Pomeranian Medical University, Powst. Wlkp. 72, 70-111 Szczecin, Poland; Mirza.Baig@med.uni-tuebingen.de (M.S.B.); anna.pilutin@pum.edu.pl (A.P.); barbara.wiszniewska@pum.edu.pl (B.W.); 2Department of Biochemistry and Medical Chemistry, Pomeranian Medical University in Szczecin, Powst. Wlkp. 72, 70-111 Szczecin, Poland; chrissaf@mp.pl (K.S.); irena.bosiacka@pum.edu.pl (I.B.-B.); 3Department of Nephrology, Transplantology and Internal Medicine Pomeranian Medical University, Powst. Wlkp. 72, 70-111 Szczecin, Poland; askodom@poczta.onet.pl

**Keywords:** kidney, finasteride, androgen receptor, apoptosis, proliferation, junction proteins, IL-6, lymphocytes infiltration, renal fibrosis

## Abstract

In the pharmacological treatment of prostate cancer, benign prostatic hyperplasia and androgenetic alopecia finasteride is commonly used. This drug inhibits 5α-reductase type 2, which is why finasteride affects androgen homeostasis, since testosterone (T) cannot be reduced to dihydrotestosterone (DHT). As studies on sex-related renal injuries suggest a high probability of androgen-induced renal dysfunction, the aim of this study was to determine the potential harmful effects of finasteride on the kidneys of rats. The study was performed on sexually mature male Wistar rats given finasteride. Histological sections of the kidneys were used for immunohistochemical visualization of the androgen receptor (AR), junctional proteins (occluding (Occ); E-cad, N-cad, E-/N-cadherin; β-cat, β-catenin; connexin 43 (Cx43)), proliferating cell nuclear antigen (PCNA), IL-6, and lymphocyte markers (CD3 for T cell, CD19 for B cell). The TUNEL method was used for cell apoptosis identification, and picro sirius red staining was used to assess collagen fibers thickness. The levels of T, DHT and estradiol (E2) were determined in blood serum. It was shown that finasteride treatment affected steroid hormone homeostasis, altered the expression of AR and intracellular junction proteins, changed the ratio between cell apoptosis and proliferation, and caused lymphocyte infiltration and an increase of IL-6. The thickening of collagen fibers was observed as tubular fibrosis and glomerulosclerosis. Summarizing, finasteride-induced hormonal imbalance impaired the morphology (i.e., dysplastic glomeruli, swollen proximal convoluted tubules) and physiology (changed level of detected proteins/markers expression) of the kidneys. Therefore, it is suggested that patients with renal dysfunction or following renal transplantation, with androgen or antiandrogen supplementation, should be under special control and covered by extended diagnostics, because the adverse negative effect of DHT deficiency on the progression of kidney disease cannot be ignored.

## 1. Introduction

Finasteride is a steroidal inhibitor of 5α-reductase type 2 (*5α-red2*), which prevents irreversible reduction of T (testosterone) into DHT (dihydrotestosterone) [1,2]. It is commonly used in the treatment of prostate cancer, benign prostatic hyperplasia (BPH), and androgenic alopecia (AGA) [3,4,5]. However, as indicated by recent studies on androgen deprivation therapies (ADTs), it is possible that finasteride has an adverse effect on kidneys. A multiple cohort study with newly diagnosed non-metastatic prostate cancer showed that the use of ADT increased the risk of acute kidney injury (AKI) [6]. Furthermore, ADT in patients with prostate cancer may antagonize the vasodilating effects of testosterone on renal vessels [7], while estrogen deficiency can negatively affect renal tubular function [8].

The influence of sex hormones on the kidney is associated with the close connection between urinary and reproductive systems [9]. For this reason, the proper development and correct physiology of the urogenital (UG) system significantly depends on androgen-estrogen homeostasis [10,11]. Evidence of this can be seen in the expression of the androgen (AR) and estrogen (ER) receptors in different parts of the nephron and tubular duct system in the kidney [12,13,14].

Many studies provide evidence for androgen-induced promotion of renal injury [15,16,17]. For example, male laboratory rats of most strains have a high susceptibility to the development of proteinuria and glomerulosclerosis, as well as the development of chronic allograft nephropathy, while females and castrated males seem to be more resistant to these abnormalities [15,16,18,19,20]. Sex differences have also been shown in the progression of hypertension and renal disease in animals and humans [21,22,23,24,25,26]. An association has been shown between the male sex and a more rapid progression of kidney diseases irrespective of blood pressure and cholesterol levels [27]. Moreover, men exhibit a more rapid age-related decline in renal function than women, and some renal diseases are clearly sex-dependent [25,28]. Membranous glomerulonephritis, IgA nephropathy, and polycystic kidney diseases are highly correlated with the male sex, where for instance Goodpasture’s syndrome is three times more frequent in men than in women [22,29]. A clinical study showed that strong AR-positive signals and a relatively higher level of AR transcription correlated with kidney stones and that men manifested a more frequent presence of stones than women [13]. In addition, testosterone is widely believed to promote progressive renal damage in males [18,19,20,25]. In research conducted on piglets, an intra-arterial infusion of testosterone dilated not only coronary, mesenteric, and iliac circulation but also renal circulation and the mechanism of this response involved the release of nitric oxide [7].

Some studies describe a correlation between androgen/AR level and the expression of intercellular junctions, i.e., tight, adherens, and gap junctions (TJs, AJs, and GJs), whose cooperation is crucial for the proper physiology of the kidney [30,31,32,33,34]. The apically located TJs, formed among others by occludin (Occ) [35], function as the primary barrier against fluids, electrolytes, macromolecules, and pathogens and are responsible for restricting paracellular diffusion into the intracellular compartment or so-called “gate and fence function” [36,37]. In contrast, AJs, which are associated for example with E-cadherin, N-cadherin, and β-catenin [38], are crucial for the maintenance of renal epithelial cell polarity and integrity, and it is suggested that they control the function of the TJs [30,31,32] or can affect renal fibrosis [39]. Connexons of GJs are channels made up by connexins (Cxs); there are many isoforms of these proteins and one of them presented in the kidney is connexin 43 (Cx43) [40]. Gap junctions are responsible for the transmission of intracellular signals and direct exchange of small molecules and ions such as Na^+^ and K^+^ [41]. In hormone-dependent tissue, such as the kidney, nuclear hormone receptors such as AR, and their ligands, such as androgens, can target the expression, physiology (modification, molecular interactions, and stability), and localization of junction proteins [33,34]. Androgens/AR expression influence not just these mentioned proteins associated with anchoring junctions. Administration of dihydrotestosterone has been shown to counteract a high castration-induced increase in Cx43 mRNA and protein in the prostate ventral lobe [42]. The authors of that study underline the physiological role of gap junctions and androgens in the regulation of prostate homeostasis, as a link to a better understanding of androgen-dependent prostate carcinogenesis.

Given the above, the aim of the study was to evaluate the effects of 4–5-month postnatal exposure to finasteride (*5α-red2* inhibitor) on blood sex hormone (T, DHT, and E2) levels and the androgen receptor (AR) and junction protein (E-cad, N-cad, β-cat, Occ, and Cx43) expression in the kidney. Secondly, we aimed to check if DHT deficiency is a stress factor to kidney cells, possibly leading to change in the apoptosis/proliferation index and IL-6 expression causing lymphocyte infiltration or a change in kidney morphology.

## 2. Materials and Methods

### 2.1. Animals

Sexually mature male Wistar rats (three months old, *n* = 10) were individually housed in cages in a 12/12 h light/dark cycle and given food and water ad libitum. The animals were randomly divided into a control (*n* = 5) and an experimental (finasteride-treated rats; *n* = 5) groups. Finasteride (Proscar^®^, MSD, Cramlington, UK) was given once per day in the morning for 4–5 months as a small pellet of finasteride powder (5 mg/kg bw) placed in bread to each experimental male rat. The animals willingly ate the pellets from the hand of the person performing the experiments. Once a week the animals were weighed and the finasteride doses adjusted. The dose of finasteride was the same as in our previous investigation [43,44,45] and as described by others [46,47]. After the experiment period, the rats were terminated with thiopental (Biochemie GmbH, Austria) at 120 mg/kg bw intraperitoneally [44,45]. However, thiopental is believed to modify sex hormone levels, and the possible changes in these parameters were the same in both groups of animals, so correlation in the results between one group (Control) and the other (Fin) was possible. The study was approved by the Local Ethics Committee for Scientific Experiments on Animals in Szczecin (Poland), approval number: 23/2010.

### 2.2. Hormone Assays

The procedure of measurement of T and DHT in blood plasma is the same as in the previously published work [44]. Blood was obtained from the rat heart using EDTA as an anticoagulant, cooled and centrifuged for 15 min at 1000 *g* at 8 °C. The collected plasma was stored at −80 °C for further hormone analysis. A standard sandwich ELISA assay was performed on the plasma using a rat specific T and DHT ImmunoAssay System kit (CUSABIO; CBS-E05100r and CBS-E07879r), according to the manufacturer’s instructions. To measure the hormone levels, an Asys UVM 340 microplate reader (Asys Hitech Gmbh, Austria) was used. The T and DHT protein concentration was normalized to total protein levels as measured by a BCA kit (Pierce, USA) using bovine albumin as a standard. The 17β-estradiol (E2) concentration was evaluated by ELFA immunofluorescence (Enzyme Linked Fluorescent Assay) on a MINI VIDAS analyzed (Bio Merieux, France).

### 2.3. Immunohistochemistry (IHC)

The kidneys were fixed in 4% buffered formalin, then washed with absolute ethanol (3 times over 3 h), absolute ethanol with xylene (1:1) (twice over 1 h), and xylene (3 times over 20 min). Then, following 3 h of saturation of the tissues with liquid paraffin, the samples were embedded in paraffin blocks. Using a microtome (Microm HM340E), 3–5 µm serial sections were taken and placed on polysine histological slides (Thermo Scientific, UK; cat. no. J2800AMNZ). The sections of the kidneys were deparaffinized in xylene and rehydrated in decreasing concentrations of ethanol, and then used for immunohistochemical staining. In order to expose the epitopes, the sections were boiled twice in a microwave oven (700 W for 4 min and 3 min) in 10 mM citrate buffer (pH 6.0). Once cooled and washed with PBS, the endogenous peroxidase was blocked by a 3% solution of perhydrol in methanol, and the slides were then incubated for 60 min at room temperature (RT) with primary antibodies. To visualize the antigen-antibody complex, a Dako LSAB+System-HRP was used (DakoCytomation, Code K0679, Dako Inc, Carpinteria, CA, USA) based on the reaction of avidin-biotin-horseradish peroxidase with DAB as a chromogen, according to the included staining procedure instructions. Sections were washed in distilled water and counterstained with hematoxylin (apart from the slides covered by anti-AR antibody). For a negative control, specimens were processed in the absence of primary antibodies. Positive staining was defined microscopically (Leica DM5000B, Wetzlar, Germany) by visual identification of brown pigmentation [44]. Primary antibody list: AR (Dako Cytomation, M3562, final dilution 1:500), PCNA (SantaCruz Biotechnology, sc-25280, 1:250), IL-6 (sc-130326, 1:250); adherens junction: E-cadherin (sc-7870, 1:100), N-cadherin (sc-7939, 1:100), β-catenin (sc-7199, 1:100); tight junction: occludin (sc-5562, 1:100); gap junctions: connexin 43 (sc-59949, 1:500); lymphocytes: CD3 (sc-70626, 1:250) and CD19 (sc-85000-R, 1:250).

Five microphotographs were taken from each kidney slide (Control, *n* = 5; Fin, *n* = 5), at the same objective magnification (x40), to provide equal analysis area, for a total of 50 analyzed photomicrographs from each immunohistochemistry (IHC) reaction. The IHC reactions were evaluated in the renal corpuscle (RC), the proximal convoluted tubule (PCT), and the distal convoluted tubule (DCT). The samples stained by the IHC procedure were independently examined by two experienced histologists (double-blind analysis to avoid suggesting and falsifying the results). AR-positive cells (regardless of localization in cell and intensity of IHC reaction) were counted separately in PCT, DCT, and RC and were given as a percentage of the total cell number in each mentioned structure of nephron. The portions of the nephron (PCT, DCT, and RC) positive for junctional protein were also counted separately, and according to the intensity of the brown color of the IHC reaction, the mentioned structures were given a score of negative (0), weak (1), moderate (2), strong (3), or very strong (4).

### 2.4. Collagen Fiber Visualization and Validation

To detect the different types of collagen in tissues, we used Sirius Red staining (Direct Red 80 Sigma Aldrich—0.1% of Sirius Red in saturated aqueous picric acid). Deparaffinized sections of formalin fixed kidney were stained as described by Junqueira et al. [48]. Collagen fibers were visible in the polarized light microscope by yellowish-orange birefringence (thick, type I collagen) or greenish-yellow birefringence (thin, type III collagen) color. The color and intensity of birefringence depended on differences in the pattern of physical aggregation and the thickness of the collagen fibers. Thin collagen fibers exhibit green to greenish-yellow polarizing colors, whereas thick fibers exhibit yellowish-orange, orange, and red colors [48].

The thickness of the collagen fibers was measured in kidney sections (5 microphotographs were taken using 20× magnification—to provide equal analysis area) from each control (*n* = 5; 25 section areas analyzed) and experimental (*n* = 5; 25 section areas analyzed) group of rats, in the area between convoluted tubules (in the interstitial region of cortex) and within the renal corpuscle. The percentage of area occupied by collagen fibers in comparison to the entire area of slide (100%) was calculated in 5 photomicrographs from each animal. Morphometric valuations (thickness of fibers and area occupied by collagen) were made using the LAS v4.4 Core Analysis software (Leica Microsystems CMS GmbH, Switzerland). 

### 2.5. Apoptosis In Situ Detection

Analysis was performed using a TACS^®^ 2 TdT-DAB In Situ Apoptosis Detection Kit (TRAVIGEN^®^ Inc., cat. no. 4810-30-K, Gaithersburg, Maryland, USA). The detailed procedure according to the included staining procedure instructions is also described in the previously published work [43]. Deparaffinized sections of formalin-fixed kidneys were incubated with proteinase K solution (15 min at RT), rinsed in PBS twice, and incubated for 5 min with a mixture of methanol (POCH, Poland) and 30% hydrogen peroxide (Sigma, Poland) (45 mL CH_3_OH + 5mL H_2_O_2_). The slides were then covered with TdT Labeling Buffer for 5 min and incubated with Labeling Reaction Mix (mixture of TdT dNTP Mix, TdT Enzyme, Mn^2+^, TdT Labeling Buffer) for one hour. The negative control was incubated with Labeling Reaction Mix omitting the TdT Enzyme. After this time, the reaction was halted with TdT Stop Buffer, and the slides were rinsed in PBS and labeled with streptavidin conjugated with horseradish peroxidase. To visualize the effect of the reaction (places of DNA split), 3,3’-diaminobenzidine was added. Positive staining was defined microscopically (Leica DM5000B, Germany) through visual identification of brown pigmentation of the cell nucleus.

#### Validation of TUNEL-Positive (Apoptotic Cell) and PCNA-Positive (Proliferating Cell)

From each kidney slide (Control, *n* = 5; Fin, *n* = 5), 5 microphotographs were taken (objective magnification: x10) to provide equal analysis area, wherein the TUNEL-positive and PCNA-positive cells were counted separately in the proximal (PCT) and distal (DCT) convoluted tubules. A total of 50 photomicrographs were analyzed. The TUNEL-positive or PCNA-positive cells per tubule were counted, and the percentage of the tubular cross-section containing positive cells was evaluated, as shown in Figure 1.

### 2.6. Statistical Analysis

Statistical analysis of the obtained results was conducted using Statistica 10 software (Statsoft, Krakow, Poland). The results are shown as arithmetic mean (AM) ± standard deviation (±SD). The distribution of variables was evaluated using Shapiro–Wilk test. If the results deviated from normal distribution, a non-parametric Mann–Whitney U-test was used for comparison between finasteride-treated and control rats. If the distribution was normal, a Student’s t-test was applied. The probability *p* ≤ 0.05 was considered statistically significant.

## 3. Results

### 3.1. Sex Hormone Levels

Statistically significantly decreasing concentrations of both androgens (T and DHT) in the finasteride-treated rat blood serum was evident, along with the level of 17β-estradiol (E2—not statistically significant) (Table 1).

### 3.2. Androgen Receptor Expression

According to the immunohistochemical reaction, AR expression in the DHT-deficient rats showed a downregulation in the cortical region of the kidney (Table 2, Figure 2).

In control kidneys, about 70% of cells in PCT and 90% in DCT epithelial cells expressed the androgen receptor in comparison to finasteride-treated rats where only 16% or 5% of cells were AR-positive (in PCT and DCT, respectively), and these differences were statistically significant (Table 2). In both groups of rats, only a few cells in the renal corpuscle possess this receptor. In control rats’ kidneys, epithelial cells have a cytoplasmic presence of AR, mainly around the nucleus or in the apical region of cells (Figure 2A,a; red and black arrows). The expression of AR in DHT-deficient rats was not only reduced (Table 2) but also underwent translocation from cytoplasm into the cell nucleus (Figure 2B,b; green arrows).

### 3.3. Junctional Protein Expression

In the kidneys of the rats with DHT deficiency, changes in the expression adherens and gap junction proteins were noticed (Table 3; Appendix A). Only occludin from the studied junctional protein did not change its expression after finasteride treatment (Table 3; Appendix A). In finsteride-treated rats, the level of gap junction protein, Cx43, strongly increased in the renal corpuscle and in the PCT; in distal convoluted tubules, the level of Cx43 seemed to be unchanged (Table 3; Appendix A), but statistical analysis showed an increase at the borderline. The transmembrane (E-cad, N-cad) and cytoplasmic (β-cat) proteins that constituted adherens junctions in DHT-deficient rats’ kidneys underwent a statistically significant reduction in RC (apart from β-cat), PCT, and DCT (Table 3; Appendix A).

### 3.4. Apoptosis/Proliferation Ratio

This test was considered positive by the presence of a color (brown) reaction in at least one nucleus in the tubule (Figure 3; arrows), and positive signals of both reactions (TUNEL and PCNA reactivity) were most frequent in DCT (Figure 3, blue arrows) than PCT (Figure 3, red arrows). Moreover, positive results are expressed as a percentage of proximal or distal convoluted tubules with a stained nucleus. The number of apoptotic/proliferating nuclei were analyzed at each proximal and distal tubule in the samples separately. In Table 4, the results are presented as a percentage of nuclei with features of apoptosis or proliferation among all the nuclei in one cross-sectioned tubule.

In the Fin group, PCNA-positive cells (Figure 3a,b) per DCT accounted about 17%, which was statistically significant (Table 4) more than twice as frequently as was the control (8%). In the PCT, no significant statistical relationship was shown, although the percentage of PCNA-positive cells was about two times higher (7%) in comparison to control rats (4%) (Table 4).

Similarly, TUNEL-positive cells (Figure 3A,B) in the PCD and DCT of Fin group were about one and a half times more frequent than the control; however, statistical values were borderline significant and only for DCT (Table 4).

### 3.5. Lymphocytes T and B Specific Markers and IL-6 Expression

Positive immunohistochemical reaction for CD3 (T cell) and CD19 (B cell) was especially noticed in huge concentrations around atrophic, pathologically changed tubules in the finasteride-treated rats (insertion in Figure 4B; green frame, red arrows) or in occasionally observed glandular structures formed by lymphocytes (insertion in Figure 4b; red arrows).

In finasteride-treated rats, IL-6 was expressed around the pathologically changed structures such as the above-mentioned atrophic pathologically changed tubules (insertion in Figure 5B; red frame, red arrows). The expression was also noticeable in renal corpuscles (RC) and epithelial cells of the nephron (Figure 5A,B; red arrows).

### 3.6. Renal Fibrosis (Collagen Fiber Thickening)

Two types of collagen fibers were identified by a polarized microscope, in bright yellow/orange (type I collagen) and green (type III collagen) (Figure 6 and Figure 7).

The thickness of type I collagen fibers was statistically significantly higher (Table 5) in the interstitial region of the kidney cortex in the finasteride-treated rats, especially around renal convoluted tubules, causing histological features of tubulosclerosis (Figure 6B, black arrows). Additionally, an increase amount of collagen fibers was also observed in glomeruli (Figure 6b; black arrows), but the change in average thickness of type I fibers was not statistically significant (Table 5). Otherwise, an increase in the amount of type III collagen fibers was mostly observed in glomeruli (Table 5), leading to glomerulosclerosis in the finasteride-treated rats (Figure 6b, black arrows). The percentage contents of collagen was over two times higher in the kidney cortex of finasteride-treated rats than in control animals (Table 5).

Dysplastic glomeruli, swollen tubules (mainly PCT), and general fibrosis were frequently observed in the experimental finasteride-treated rats (Figure 7A–C).

## 4. Discussion

In our investigation, we observed that long-lasting (3–4 months) finasteride treatment of adult male rats caused a decline in sex hormone levels, firstly with DHT, then also T and E2, although the level of estradiol did not change statistically significantly. These results are inconsistent with those presented in the available literature. The website of Archived Drug Label [50] states that, after the finasteride administration, “mean circulating levels of testosterone and estradiol were increased by approximately 15%,” which is consistent with the research [51,52]. Moreover, the level of E2 increase is probably due to the aromatization of bioavailable T. On the other hand, Antus et al. [15] did not observe that finasteride treatment significantly influenced serum testosterone levels. These discrepancies in all the aforementioned results may be due to the dose and the length of finasteride administration; for example, Antus et al. [15] used doses of 25 mg of finasteride/kg bw every second day over 20 weeks, and animals after kidney transplantation were also given the antibiotics and immunosuppressive drugs.

Because finasteride affects the hormonal homeostasis, it could be compared to endocrine disrupting chemicals (EDC) with estrogenic/antiandrogenic activity. Antus et al. [15] showed that finasteride and flutamide (nonsteroidal antiandrogen, the antagonist of the androgen receptor) acted as EDC and improved long-term allograft outcome after kidney transplantation. Thus, the physiology of kidney is mediated by androgen receptors (AR) localized in the cells of most parts of the nephron. In the cortex of the kidney, AR is located in various structures, predominantly in proximal and distal convoluted tubules (PCT and DCT) and shows focal expression in the parietal layer of Bowman’s capsule [53,54,55,56,57].

There is one common androgen receptor for the different androgens (T, DHT, etc.), and its function depends on the concentration of the hormone [58]; for instance, downregulation of the receptor was observed in prostate cancer after the deprivation of androgens [59]. Testosterone and dihydrotestosterone may influence the expression of other genes differently as a consequence of the different binding affinities of the receptor. In peripheral target tissues, DHT would be necessary for androgen action, whereas in tissues in which T concentration is high (testes, Wolffian ducts), DHT formation may not be essential for virilization [60]. The androgen receptor is involved in the initiation and progression of various types of cancers (bladder, kidney, lung, breast, and liver, but not the prostate, where it acts as a suppressor) [61]. Among the various types of tumors within one tissue/organ, differences in the level of AR expression could be found. Bass et al. [62] showed a lower expression of AR in prostate cancer cells in comparison to benign prostate cells. Renal cell carcinoma (RCC) is hormone-dependent: AR was found in 15% of patients with RCC and inversely correlated with the histopathological stage (27% of cells were AR-positive in pT1 tumors, only 4% cells were AR-positive in pT3 tumors) and inversely correlated with the nuclear grade of receptor expression. Univariate analysis showed longer disease survival in patients with AR-positive tumors compared to patients with AR-negative tumors [57]. In our experiment, we observed a decreased level of expression of AR in the cortical region (RC, PCT, and DCT) of the kidney in the finasteride-treated rats that could be related to altered serum androgens (T and DHT) levels. In a study by Chang et al. [61], AR also has a possible effect on the progression of renal cell carcinoma, because the cancer cells display a lower expression of the receptor. In contrast, strong AR-positive signals and a relatively higher level of AR transcript correlate with kidney stones [13]. Enhanced AR signaling plays a promoter role in the early stages of calcium oxalate (CaOx) crystal formation by increasing oxalate biosynthesis in the kidneys. Therefore, the targeting of AR could be developed as a potential therapy to battle CaOx crystal-related kidney stone disease [13]. However, it was also documented that estrogens influenced urinary oxalate excretion, crystal deposition, and calcium content in renal tissue in the oophorectomized rats, and these parameters were inhibited by female sex hormone supplementation [63]. Basically, androgen-estrogen homeostasis is a key event in organ physiology, for example in the context of the heterodimerization of ER with AR and the formation of an active transcriptional factor [64], as well as the existence of cytoplasmic, nuclear, and plasmalemma-associated steroid receptors [65,66,67,68,69,70]. For instance, in kidneys without RCC the location of AR is cytoplasmic, while in renal cell carcinoma it is nuclear [71]. The alteration of androgen homeostasis by finasteride in our experimental rats resulted also in AR translocation from cell cytoplasm to the nucleus. Zhu et al. [71] indicated that this transformation probably happens due to some hereditary agent or environmental exposure, and in this case finasteride could probably play a role as the environmental factor. Moreover, AR has been described as a factor responsible for the deterioration of renal function and can be predisposed to be a target for chronic kidney disease (CKD) treatment [72].

In our experiment, the decrease of AR expression finally lead to histopathological changes in the kidney cortex, such as fibrosis and infiltration of mononuclear cells that were also observed in CKD by Guan et al. [72]. Supplementation with testosterone caused histological changes in animal kidneys (glomerulosclerosis, tubulointerstitial fibrosis, tubular atrophy, and infiltration by mononuclear cells) and increased urinary protein excretion (proteinuria) [15,73]. Antus et al. [15] showed an opposite effect for flutamide (the antiandrogen) and finasteride treatment, which caused reductions in glomerulosclerosis, CD5^+^ T lymphocytes, and the number of monocytes/macrophages; these infiltrations by immune cells were associated with decreased TGF-β, PDGF-A, and/or PDGF-B expression. In contrast, in our study, the finasteride treatment increased glomerulosclerosis, tubulointerstitial fibrosis, and the infiltration of mononuclear cells. These differences could be due to the dose of finasteride and the duration of treatment (Antus et al. used doses of 25 mg/kg bw every second day over 20 weeks; however, this subcutaneous finasteride injection perhaps leads to bioavailability as high as that following high oral dose used in our experiment) or because of the treatment of the animals with antibiotics and immunosuppressive drugs to prevent transplanted kidney rejection or just because of compound flutamide/finasteride action [15]. In the context of mononuclear cell infiltration, we observed an increase in the number of CD3^+^ and CD19^+^ cells mainly around atrophic pathologically altered tubules in the finasteride-treated rats. It has been documented that T cells can express TGF-β and directly induce adjacent tubular epithelial cells transformation to proliferating fibroblasts that migrate across the tubular basement membrane, producing fibrotic lesions within the renal interstitium [74]. In our experiment, the rise in immune cell numbers was positively correlated with the intensification of IL-6 expression, not only in the region of mentioned pathologically altered tubules but also in the area of the renal glomeruli and their epithelial cells. Araneo et al. [75] documented that DHT exerts a depressive influence on IL-4 and IL-5 production but not on IL-2. Perhaps in our study the increase in IL-6 expression was the result of a mechanism similar to one that controls IL-2. T cells and macrophages possess testosterone-binding sites [76], and macrophages demonstrate 5α-reductase activity [77]. Thus, the direct effects of androgens and their influence on immune cells could have contributed to the number of lymphocytes T and B observed in our study. Sex hormones not only control renal hemodynamics, mesangial cell proliferation, and extracellular matrix metabolism, but also affect the synthesis and release of some cytokines and other growth factors, which in turn are capable of altering the progression of renal diseases [29]. Humoral and cellular immune responses are considered to be regulated by sex hormones [12,77]. Therefore, the impaired sex hormone ratio in our finasteride-treated rats could exactly reflect the immune cell response and proinflammatory factor concentration.

Androgens may modulate the synthesis/release of cytokines and growth factors that exert strong profibrogenic effects and stimulate mesangial cell proliferation [12]. Testosterone supplementation resulted in a rise in vascular smooth muscle cell proliferation in the kidney graft [15]. An increase in epithelial cell turnover (two times higher but statistically significant only in DCT) was observed in our finasteride-treated rats with lower androgen levels (T and DHT). In a study by Antus et al. [15], the animals were receiving antibiotics and immunosuppressive drugs, so the conditions of the experiment were totally different. In our study, it can be hypothesized that the inhibition of the 5alpha-reductase leads to a decrease in blood testosterone, because free T could be more willingly and quickly aromatized to estradiol, a strong mitotic inducer; therefore, we observed the enhanced proliferative activity of the cells. The organ reaction may have been due to different mechanisms connected with each other. For example, finasteride has been shown to affect the expression of proteins belonging to the Bcl-2 family [78], leading to the suppression of the genes encoding insulin-like growth factor 1 (IGF-1) and the receptor for IGF-1, thus inhibiting cell proliferation [79]. The gene encoding IGF-1 is androgen-dependent, whereas the gene encoding the IGF-1 receptor is a target for estrogens [80]. Therefore, the modulation of these genes is a good example of the cooperative action of sexual steroid hormones by their receptors (in the meaning of the transcriptional factors).

Androgens have been shown to increase proapoptotic signaling [81]. Testosterone-induced apoptosis in proximal tubule cells [82] involves the activation of inflammatory cytokines such as c-Jun amino terminal kinase (JNK) and can be blocked by the AR antagonist – flutamide, which reduces JNK phosphorylation [83]. Accordingly, there is a cooperative action of androgens with estrogens in cell turnover. In our study, the androgen-deprived rats display an elevated apoptosis index, mainly in DCT. A study performed on patients indicated that finasteride treatment caused prostate involution through a combination of atrophy and cell death [84]. A study carried out on dogs with benign prostatic hyperplasia is in agreement with the above data, and showed that finasteride-induced prostatic involution appears to be via apoptosis rather than necrosis, and the percentage of apoptotic cells depends on the duration of treatment [85]. The finasteride initiation of cell death could be different in the various neoplastic (cancerous and benign) cells [62].

The change we observed in the apoptotic/proliferating ratio could be triggered by altered cell communication through gap junctions. The consequence of the close association of development of the urinary with reproductive systems, in kidney Cx43 expression could also be androgen-dependent, as in the genital tract [33,86,87]. DHT-deficient rats revealed an increase in Cx43 expression in renal corpuscles and PCT (which revealed pathological changes—expanded/swelled lumen), and this resulted in kidney changes such as alterations in the exchange of small molecules or ions (Na^+^ and K^+^), juxtaglomerular apparatus calcium signaling or tubular purinergic signaling gated by connexons [40,88]. Gap junctions play a role in the renin-angiotensin system, tubuloglomerular feedback, and salt and water reabsorption and, as a consequence, help regulate blood pressure, and they could be involved in hypertension and diabetes [40]. Additionally, proximal sodium reabsorption and intraglomerular pressure (modulating afferent/efferent arteriolar tonus via angiotensin II and endothelin) are controlled by androgens [89]. The DHT deficiency in our study also caused a change in adherent junction protein expression; this modification was spread within the cells of PCT, which had signs of lumen dilatation, but this expression was also changed in RC and DCT. This could be related to the establishment that AJs are essential for the preservation of renal epithelial cell polarity and integrity, and control the function of the TJs and gap junctions as well [30,31,32], but we did not observe any changes of occludin expression after finasteride treatment.

The observed altered renal cell turnover process could also have been due to fibrosis. According to Rastaldi et al. [90] tubular epithelial cells (TEC), via transdifferentiation to a mesenchymal phenotype, can produce extracellular matrix (ECM) proteins in human disease and directly intervene in fibrotic processes. Moreover, they showed that the TEC were positive for PCNA. It was documented that adherent junction proteins also affect renal fibrosis [39] as a result of epithelial-mesenchymal transition [91]. The steroid/nuclear receptor regulated the dynamics of occluding and anchoring junctions [33,34], which is why the changed expression level of the studied adherent junction proteins (E-cad, N-cad, and β-cat) in the finasteride-treated rats might impair kidney morphology. Clinical observation on patients with prostate cancer showed a positive correlation between AR and β-catenin expression in the tumorigenic cells, and an activated Wnt/β-catenin pathway and AR expression in prostate cancer strongly correlated with the metastasis and aggressiveness of the tumors [92]. The cancer-related aspects of Wnt/β-catenin signaling was also shown on other hormone-related male reproductive organ such as the epididymis [93]. The critical role of the Wnt pathway in nephrogenesis is well established, and research has shown its involvement in many adult kidney diseases, including ischemic kidney injury, glomerular diseases, diabetic nephropathy, interstitial fibrosis, and cystic kidney diseases. Overall, activation of the Wnt pathway is deleterious to many chronic diseases of the kidney, contributing to the maintenance of cells in an activated state. In addition, the Wnt pathway is activated during repair and regeneration in acute ischemic injury [94].

## 5. Conclusions

The finasteride treatment of adult male rats led to a decrease in androgen receptor expression and its cellular translocation within the kidney cortex.The pathomorphological changes (glomerulosclerosis, tubulosclerosis, dysplastic glomeruli, and tubules with lumen dilatation) in rats’ kidneys with disturbed steroid hormone imbalance were associated with the diminished expression of intracellular junctional proteins.The changed apoptotic/proliferating ratio of nephron cells and the increase in the numberof lymphocytes in the area of pathologically altered convoluted tubules were accompaniedby impaired androgen/estrogen homeostasis.

Although studies of exogenous DHT supplementation in animals previously receiving finasteride have not been performed, it can be suggested that these described evidences from the animal model of experiment could indicate that the patients with renal dysfunction or following renal transplantation with androgen supplementation or with pharmacologically (i.e., by finasteride) induced DHT deficiency should be under special control and covered by extended diagnostics, due to the potential adverse effect of DHT deficiency on the progression of kidney disease.

## Figures and Tables

**Figure 1 ijerph-16-01726-f001:**
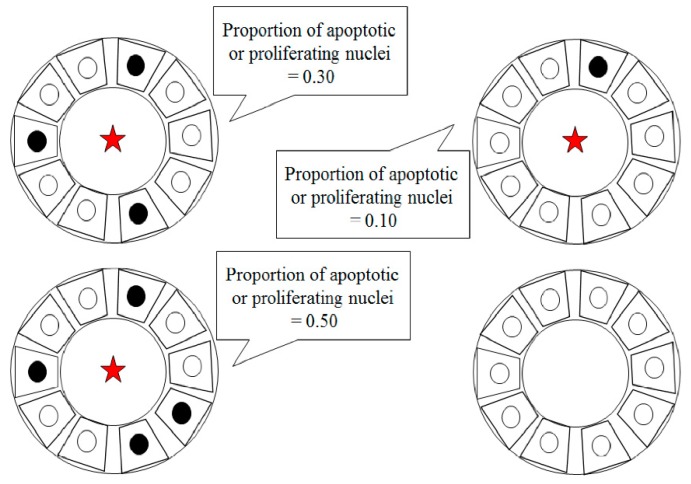
Renal tubule (PCT or DCT) cross sectional structures with positive staining to detect apoptosis or proliferation (marked with red asterisk). Calculation method for the percentage of apoptotic/proliferating cells in each tubule (black colored cell nuclei) according to Kędzierska et al., 2015 [49].

**Figure 2 ijerph-16-01726-f002:**
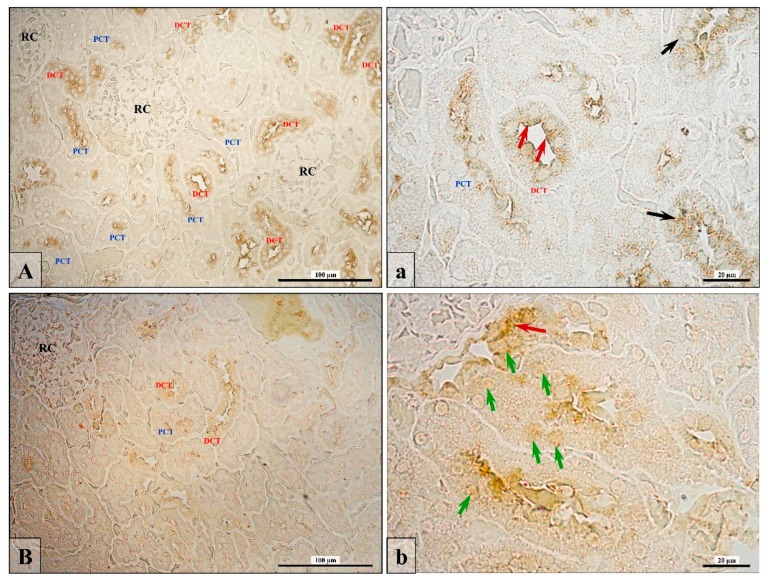
The representative microphotographs showing the expression of the androgen receptor (AR) in kidney of control (**A**,**a**) and finasteride-treated rats (**B**,**b**). In the control group, AR was most frequently expressed in DCT than PCT (**A**) and mainly in the cells’ cytoplasm around the nucleus (a; black arrows) or in the apical area of cells (a; red arrows). In rats with DHT deficiency, AR expression became very low (**B**) and mainly translocated from the apical region (b; red arrow) into the cell nucleus (b; green arrows). Scale bar from objective magnification ×40 is 100 µm (**A**,**B**), from x100 is 20 µm (**a**,**b**).

**Figure 3 ijerph-16-01726-f003:**
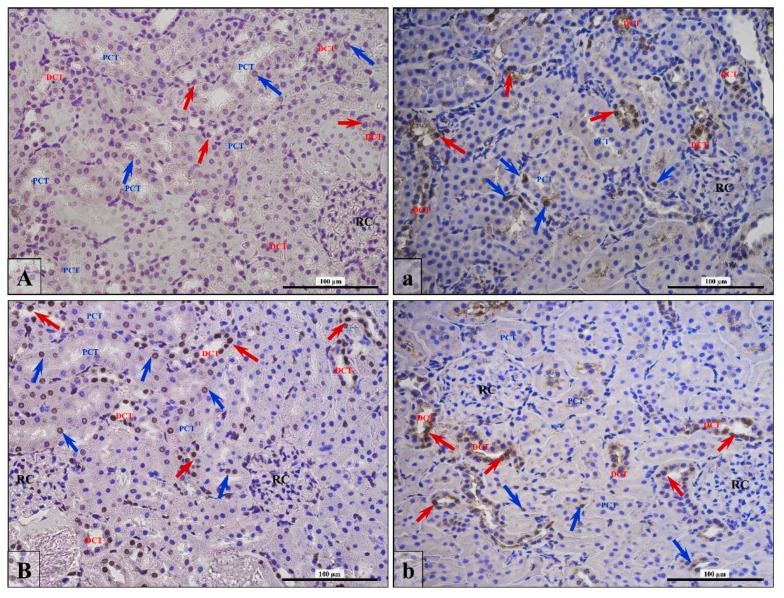
Representative microphotography showing TUNEL reaction (**A**,**B**) and PCNA-positive cells (**a**,**b**) in control (**A**,**a**) and finasteride-treated (**B**,**b**) rats. Red arrows indicate positive signaling in nuclei of PCT; blue arrows indicate positive signaling in nuclei of the DCT. Scale bar from objective magnification ×40 (**A**,**a**; **B**,**b**) is 100 µm.

**Figure 4 ijerph-16-01726-f004:**
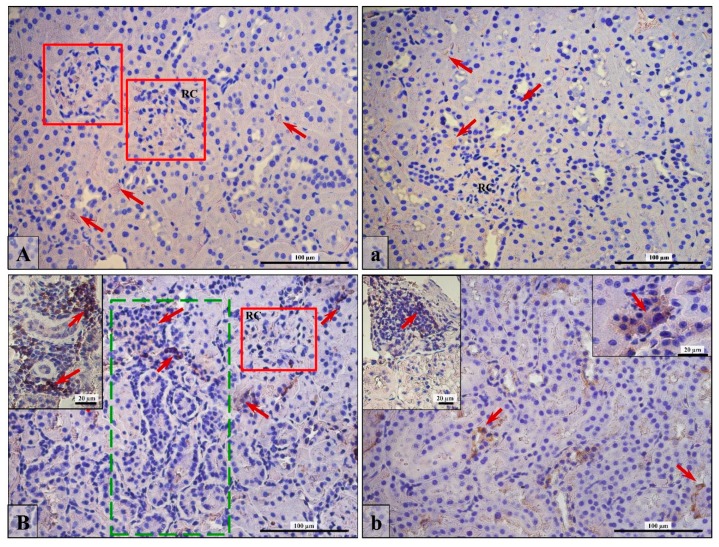
Brown pigmentation of a positive IHC reaction that indicates expression of CD3 (**A**,**B**) and CD19 (**a**,**b**) in kidney cells of Control (**A**,**a**) and Fin (**B**,**b**) groups of rats. In control rats, a kidney CD3-positive area was weakly visible in the renal corpuscle (**A; RC; red frame**), and CD19 positive cells were visible among interstitial tissue or epithelial cells (**a; red arrows**). In finasteride-treated rats, the DC3- (**B**) and CD19-positive cells (**b**) were marked by red arrows; the green frame indicates the pathologically changed area with renal convoluted tubules (**B**). Scale bars in A, a, B, and b are 100 µm; scale bars in insertion in B and b are 20 µm.

**Figure 5 ijerph-16-01726-f005:**
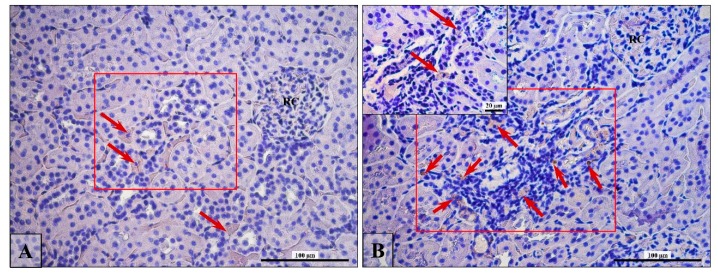
The representative microphotography showing IL-6 expression in control (**A**) and finasteride-treated (**B**) groups of rats. Red arrows indicate a positive area of immunoreactivity, red frames shown the areas with intense immunoreactivity for IL-6. Scale bars for A and B are 100 µm; scale bar in insertion in B is 20 µm.

**Figure 6 ijerph-16-01726-f006:**
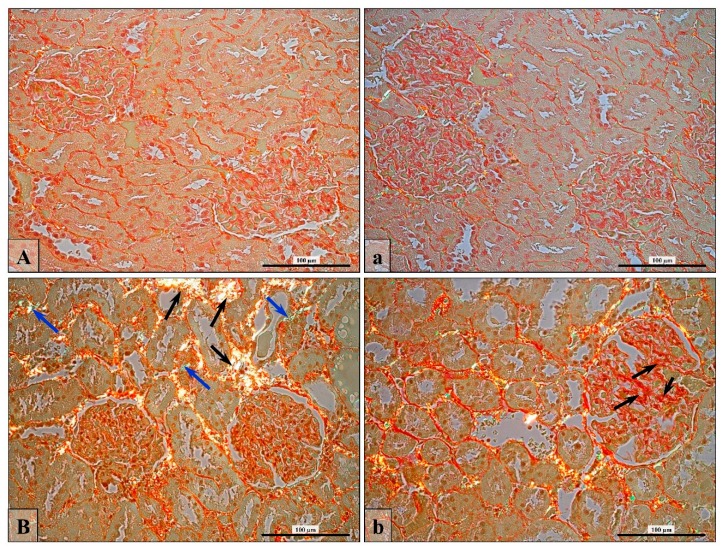
Visualization of collagen fibers type I—black arrows and type III—blue arrows) in control (**A**,**a**) and in finasteride-treated rats’ kidneys that exhibited well visible tubulointerstitial fibrosis (**B**, arrows) and glomerulosclerosis (**b**, black arrows). Sirius Red Staining, polarized microscopy. Scale bar: 100 µm.

**Figure 7 ijerph-16-01726-f007:**
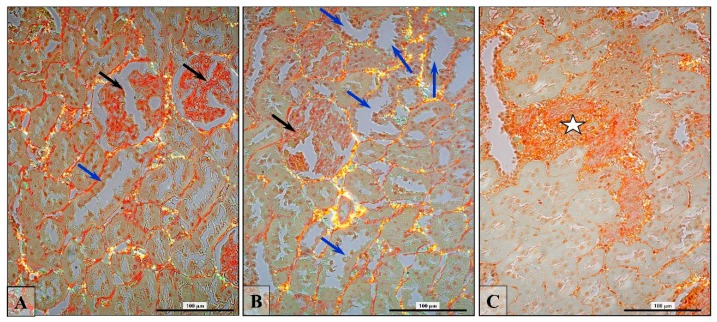
The representative microphotographs showing pathologically changed kidney of finasteride-treated rats (**A**–**C**): dysplastic glomeruli (black arrows), overloaded tubules (blue arrows), general interstitial fibrosis (asterisk). Sirius Red Staining, polarized microscopy. Scale bar: 100 µm.

**Table 1 ijerph-16-01726-t001:** Dihydrotestosterone (DHT), testosterone (T), and estradiol (E2) concentrations in control and finasteride-treated mature rat blood serum.

DHT (ng/mL)		T (ng/mL)		E2 (pg/mL)	
Control	Fin	Control	Fin	Control	Fin
0.67 ± 0.03	0.25 ± 0.09 ** vs. Control	1.53 ± 0.19	0.57 ± 0.25 ** vs. Control	56.28 ± 9.47	31.33 ± 3.63

Values are expressed as arithmetic means ± SD (*n* = 5 per each group) evaluated by the Mann–Whitney U-test. ** *p* < 0.001.

**Table 2 ijerph-16-01726-t002:** The percentage of AR-positive cells in the renal corpuscle (RC), the proximal convoluted tubule (PCT), and the distal convoluted tubule (DCT) of control and finasteride-treated (Fin) rats.

Region	AR	
	Control	Fin
RC	0.14 ± 0.12	0.06 ± 0.03
PCT	0.69 ± 0.32	0.16 ± 0.17 *
DCT	0.91 ± 0.12	0.5 ± 0.26 *

Values are expressed as arithmetic means ± SD (*n* = 5 per each group) evaluated by Student t-test. * *p* < 0.01.

**Table 3 ijerph-16-01726-t003:** The percentage of evaluated portions of nephron (RC, PCT, and DCT) according to the level (0–4) of expression of intercellular junction proteins in control and finasteride-treated rats kidney.

Occludin	Control	Fin
RC mean ± SD	100% (1) 1 ± 0	100% (1) 1 ± 0
PCT mean ± SD	18.5% (1); 81.5% (0) 0.185 ± 0.39	33.3% (1); 66.6% (0) 0.33 ± 0.48
DCT mean ± SD	58% (2); 42% (1) 1.58 ± 0.5	40% (2); 60 % (1) 1.4 ± 0.54
Connexin 43		
RC mean ± SD	100% (1) 1 ± 0	66.6% (2); 33.3% (1) 1.67 ± 0.58 ***
PCT mean ± SD	24% (3); 24% (2); 52% (1) 1.71 ± 0.84	54.5% (3); 31.8% (2); 9.2% (1); 4.5% (0) 2.36 ± 0.84 *
DCT mean ± SD	33.3% (4); 50% (3); 16.6% (2) 3.62 ± 0.71	80% (4); 20% (3) 3.8 ± 0.44
E-cadherin		
RC mean ± SD	100% (2) 2 ± 0	100% (1) 1 ± 0 ***
PCT mean ± SD	48,2% (2); 51.8% (1) 1.48 ± 0.51	7.2% (2); 17.8% (1); 75% (0) 0.32 ± 0.61 ***
DCT mean ± SD	42.8% (4); 57.2% (3) 3.43 ± 0.51	31.6% (2); 68.4% (1) 1.3 ± 0.47 ***
N-cadherin		
RC mean ± SD	100% (3) 3 ± 0	100% (1) 1 ± 0 ***
PCT mean ± SD	28,6 (1); 71.4% (0) 1.83 ± 0.41	21% (2); 52.6% (1); 26.4% (0) 0.95 ± 0.7 **
DCT mean ± SD	53.3% (3); 46.6% (2) 2.57 ± 0.5	83.3% (2); 16,6% (1) 0.28 ± 0.46 ***
β-catenin		
RC mean ± SD	100% (1) 1 ± 0	100% (1) 1 ± 0
PCT mean ± SD	45% (2); 54% (1) 1.46 ± 0.5	27.8% (1); 72.2% (0) 0.28 ± 0.45
DCT mean ± SD	62% (2); 28,6 (2); 9.4% (3) 3.33 ± 0.9	16.7% (4); 55.5% (3); 27.8% (2) 2.89 ± 0.68

Scores of intensity of IHC reaction: negative (0), weak (1), moderate (2), strong (3), and very strong (4). Values are expressed as arithmetic means ± SD (*n* = 5 per each group) evaluated by the Mann–Whitney U-test. * *p* < 0.01, ** *p* < 0.001, *** *p* < 0.0001.

**Table 4 ijerph-16-01726-t004:** The percentage of TUNEL-positive or PCNA-positive tubules (DCT or PCT) [tubule was considered as positive if the presence of a color reaction in at least one nucleus per tubule was detected] and the number of TUNEL-positive or PCNA-positive cells per one tubule (DCT or PCT) in control and finasteride-treated rats.

Apoptosis	Control	Fin
TUNEL^+^DCT	0.77 ± 0.006	0.90 ± 0.0 6 * vs. Control
TUNEL^+^PCT	0.73 ± 0.01	0.81 ± 0.14
TUNEL^+^ cells per DCT	0.18 ± 0.007	0.30 ± 0.07
TUNEL^+^ cells per PCT	0.15 ± 0.002	0.21 ± 0.043
Proliferation		
PCNA^+^DCT	0.57 ± 0.14	0.86 ± 0.02 * vs. Control
PCNA^+^PCT	0.35 ± 0.17	0.46 ± 0.16
PCNA^+^ cells per DCT	0.08 ± 0.002	0.17 ± 0.004 *** vs. Control
PCNA^+^ cells per PCT	0.04 ± 0.01	0.07 ± 0.03

Values are expressed as arithmetic means ± SD (*n* = 5 per each group) evaluated by Student t-test. * *p* < 0.01; *** *p* < 0.0001.

**Table 5 ijerph-16-01726-t005:** Average thicknesses of type I and type III collagen fibers in the interstitial cortical region and within the renal corpuscle; and percentage contents of collagen in the cortex of control and finasteride-treated rats’ kidneys.

Thickness of Collagen Fibers	Collagen Type I Fibers (µm)		Collagen Type III Fibers (µm)	
	Control	Fin	Control	Fin
	in the Interstitial Cortical Region			
Mean ± SD	1.111 ± 0.469	3.298 ± 1.760 vs. Control ***	1.561 ± 0.755	2.121 ± 1.154
	within the Renal Corpuscle			
Mean ± SD	2.542 ± 0.974	2.287 ± 0.964	0.925 ± 0.532	1.426 ± 0.569 vs. Control *
	Percentage of Area Occupied by the Collagen in Correlation to the Entire Area of the Section			
	Control		Fin	
Mean ± SD	3.63 ± 1.55		8.56 ± 0.89 **	

Values are expressed as arithmetic means ± standard deviation (*n* = 25 per each group; 5 animals in Control and Fin group; 5 microphotographs from each kidney were analyzed) evaluated by the Mann–Whitney U-test. * *p* < 0.01, ** *p* < 0.001, *** *p* < 0.0001.

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
