# Peer review of "Finasteride-Induced Inhibition of 5α-Reductase Type 2 Could Lead to Kidney Damage—Animal, Experimental Study"

_ijerph, 2019, doi:10.3390/ijerph16101726_

Round 1

Reviewer 1 Report

The manuscript was reviewed for publication in the journal. The manuscript was designed to investigate the effects of 4-5-month postnatal exposure to finasteride on blood sex hormone levels and androgen receptor and junction protein immuneexpression in the kidney and evaluate if DHT deficiency is a stress factor to kidney cells, possibly leading to change in apoptosis/proliferation index and IL-6 expression. The results obtained showed that the pathomorphological changes in finasteride-treated rats’ kidney were associated with the diminished immunoexpression of intracellular junction protein and that the changed apoptotic/proliferating ration of nephron cells and the increase of lymphocytes number in area of pathologically altered convoluted tubules were along with impaired androgen-estrogen homeostasis. 

It is the reviewer’s opinion that the manuscript is well-written and quite interesting and the results are clear. However, it appears that there are a couple of minor concerns in the manuscript.

1) There were no scales in Figure 2-7. Please add them.

2) Table 2 and 3 showed level of immunoexpression of androgen receptor and intracellular junction proteins in different parts of kidney cortex from control and finasteride-treated rats. The authors used the signs of -, +/-, +, ++, and +++ to show the level of immunoexpression, but examples of these immunoexpressions were not shown in the manuscript. The authors should add them in additional figures or supplementary figures.

3) Table 4 showed preliminary calculation of PCNA- or TUNEL-positive cell number. Why was the statistical analysis not performed?

4) Only means were shown in Table 5. Means ± standard deviation should be used?

Author Response

Dear Reviewer I,

I would like to thank you for you revision, very much. I improved my manuscript according to your suggestion and I hope that now it is fully acceptable for you.

Answer to comments:

1). The all Figures were changed, and evidently the new ones contain scale bars.

2). Additional supplementary Figure S1-S3 showing junctional protein expression (that complete the information contained in the Table 2) was added.

3) I decided to remove the Table 4 because it shown and partially duplicate the results of Table 5 (now Table 4).

4) In Table 5 (now Table 4) standard deviation was added.

Reviewer 2 Report

In the manuscript (ijerph-482097) authors could request language assistance, especially it would be instructive if authors could use scientific language. 

The discussion is too long and in my point of view can be shortened.

Author Response

Dear Reviewer II,

I would like to thank you for you revision, very much. I would like to say that manuscript was under Language Editor correction. I will sent to Journal Editing Center the certificate of the mentioned correction. I was also try to short the Discussion, but finally it was not done, because the Discussion part is already heavily compressed, and removing even the smallest part can reduce the quality of the manuscript. I improved my manuscript and the conclusions I hope that now it is fully acceptable for you.

Reviewer 3 Report

In this paper, the authors describe the effects on kidney morphology of chronic dosing with the 5-α reductase type II inhibitor finasteride in rats. They show that after 4-5 months of daily oral dosing the levels of circulating dihydrotestosterone, as well as testosterone are significantly decreased. Using immunohistochemistry they show morphological changes in kidney structure as well as evidence for changes in the expression pattern of androgen receptor, adhesion molecules, and immune response mediators. While this study certainly has its merit, there are several issues the authors need to address in order to fully support their conclusions.

A major shortcoming is that the way the IHC data is presented is not very convincing. First of all, the representative images have different levels of brightness and contrast making it hard to compare them. Furthermore, figure 2a is simply a magnification of figure 2A, which is not mentioned in the text and is therefore very misleading. The signal of the TUNEL stain and the IHC for CD3 and CD19 are all very weak and beg the question whether or not this is actual positive stain or merely artefactual. Scale bars are also missing for all figures throughout the manuscript. Why are no figures shown for the IHC of E-cad, N-cad, b-cat, occ, or Cx43? These should at least be in the supplement to allow the reader to evaluate the validity of the staining results.

Furthermore there are serious flaws in the expression analysis. For the expression of AR, E-cad, N-cad, b-cat, occ, and Cx43 a scale of -, +/-, +, ++, and +++ is used, but it is not clear how many animals and how many sections per animal were analyzed to come to the final scores reported in Tables 2 and 3. It would be much better to attempt to quantify the number of positive cells, similar to the TUNEL and PCNA analysis (although these also need some more clarification, see below). At the very least, an objective scoring matrix of 0-5 should be used to evaluate the level of staining. These values can then be averaged per animal and analyzed further using the appropriate statistical methods. Some other important questions:

-          Were the histologist blinded to the experimental groups of the sections they were scoring to avoid subjective bias?

-          How was collagen fiber thickness measured? These fibers are not uniform in thickness over their length in the sections, so it is important to know how this was standardized and how type I and type III collagen was formally discriminated from each other.

-          Why was the student T-test used to compare between groups when there are multiple groups and variables (e.g. finasteride vs control, area of interest within section, molecule stained, …)?

-          The results from Table 4, while commendable in attempting to provide a quantitative measure, are not clearly represented. Are these values the average number of positive cells per section? Or per a certain surface area? What is the standard deviation? Why is the table labeled “Preliminary” calculations? Does “degree of increase” mean “fold change”?

-          Table 5: also here the standard deviations are missing.

-          Figure 4: similar size / magnification images should be shown for control and finasteride treatment to allow an easier visual comparison between both groups.

-          Figure 5B seems to have a different magnification and have some degree of horizontal compression artefact.

Another major question pertains to the dosing schedule of finasteride. The authors state that they dosed male rats orally daily with a dose of 5 mg finasteride per kg body weight. This is however a very high dose compared to the recommended dosage in patients, which is 1-5 mg orally per day, amounting to an estimated 10 – 90 microgram finasteride per kg body weight per day. As such, the question arises whether this supratherapeutic dose of finasteride leads to off-target effects which could be responsible for the observed effects. Do the authors have evidence that the alterations in the kidney phenotype can be rescued by supplementation of DHT?

Is thiopental-induced euthanasia suitable for subsequent measurements of sex hormones? There are reports that this anesthetic has a confounding effect on different hormone levels, including testosterone (e.g. see Nazian, Proc Soc Exp Biol Med 1988). The authors state multiple times (e.g. line 194-195) that estradiol levels were lower in finasteride-treated rats, although they acknowledge that no statistical significance was reached. In that case, one has to state that the estradiol levels are not different.

In the discussion the authors attempt to reconcile their data with earlier publications, but fail to highlight some important experimental differences. The study referred to on line 289-290 looked at the effects 24h after a 1 mg oral dose, which is very different from the dosing regimen in this study. The study from Antus et al. is also referred to multiple times, but the authors fail to mention that this study only examined rats that underwent renal transplantation. In contrast to the authors’ statement on lines 351-359 the dosing regimen is rather similar, since the subcutaneous administration of finasteride likely leads to a similar bioavailability as the higher oral dose used in this study. On line 385-386 the authors state that they see an increase in testosterone levels, while the data from Table 1 clearly show the opposite.

Minor comments:

Line 22 and throughout the manuscript: “immunoexpression” does not mean what the authors think. This term is used for immune response-induced expression of proteins, while I suppose they use this term to describe the expression pattern of proteins as measured using immunohistochemical techniques.

Line 23: abbreviations should be explained in the abstract or replaced by the full name.

Line 130: I suppose 10 mM citrate buffer was used instead of 10 nM?

Figure 1 does not seem essential to the results section and would better fit in an online supplement.

Line 339 and 413: alteration instead of alternation

Author Response

Dear Reviewer III,

I would like to thank you for you detailed and very valuable revision, very much. I really appreciate your efforts and the time you have devoted to my manuscripts. Your comments are a very valuable lesson for me; a source of inspiration for subsequent experiments and advice on what mistakes do not make any more.  

Major comments:

I was changed the all Figures, it means mainly from immunohistochemical reaction. And now, I hope the results from IHC are more convincing.

The improved figures, now have similar levels of brightness and contrast, and also magnification in Control and Finasteride treated rats are the same, to make the comparison between each other easier. I hope that improved Figures meet formal requirements. Of course all microphotography include scale bars.

In the appropriate place/chapter of the manuscript was include information about the numbers  of analyzed micrographs from each kidney slide (Control, n=5; Fin, n=5), 5 microphotographs were taken (at the same objective magnification, to provide equal analysis area), which is a total of 50 analyzed photomicrographs from each IHC reaction). The analysis were curried by two independent, experienced histologists in double-blind tests to avoid suggesting and fabricating/falsifying the results.

The Figure showing adhesion molecules expression was add as supplementary data (Fig. S1-S3).

Due to the fact that the IHC reaction showing the expression of AR was localized at various sites of epithelial cells (perinuclear, apical part of cell cytoplasm or nuclear), as opposed to unambiguous (nuclear) localization in TUNEL- and PCNA-positive cells, we did not decide to count the cells (because I would have to group them into 3 categories); we have remained showing the intensity of the expression on the plus or minus scale. The same scale we used to showing expression of junctional proteins, that are presented in different part of renal epithelial cells. It was the main reason not to perform statistical analysis, that would not be as easy as in TUNEL and PCNA analysis, where the result was unambiguous (positive or negative result in the cell nucleus).

The types of collagen fibers have been distinguished by the color — yellowish-orange birefringence (thick, type I collagen) or greenish-yellow birefringence (thin, type III collagen) color [this information is included to the manuscript text]. It was measure the thickness of fibers in randomly chosen area (because as you written “fibers are not uniform in thickness over their length in the sections”).

I decided to remove the Table 4 because it shown and partially duplicate the results of Table 5 (now Table 4). In Table 5 (now Table 4) standard deviation was added.

Now, Figure 4 contains images with the same magnification, and the microphotography are free from any artefacts such as horizontal compression.

The dose of finasteride which I used in my previously and current experiment was determined based on literature data and is commonly used in animal experimental studies. I included in manuscript the information that the dose of finasteride was the same as in our previous investigation and as described by others. I would like to underline that the material (kidneys) was collected during the earlier experiment that was focused mainly on male reproductive system. I have any evidences that the alterations in the kidney phenotype can be rescued by supplementation of DHT. But I think, that is good idea to perform in the future the experiment that after the finasteride-treatment, the rats will be supplemented by dihydrotestosterone to evaluate the possible withdrawal of changes in kidney caused by used medicament.

You suggest the thiopental anesthesia is not suitable for measurements of sex hormones, according the literature data. My only defense/security may be the explanation that the used anesthetic affected hormone levels both in the control and experimental groups, therefore comparing the levels of the sex hormones in both group, the "negative effect" is eliminated. Your suggestion is a very valuable remark/advice for me in planning a new experiment, and in the future I will use a different method of animal euthanasia.

The study from Antus et al. is also referred to multiple times, but the authors fail to mention that this study only examined rats that underwent renal transplantation. – I am sorry, I don’t understand this sentence. In the second paragraph of the discussion I included information that the animals were after renal transplantation [….Antus et al. [15] showed that finasteride and flutamide (nonsteroidal antiandrogen, antagonist of androgen receptor) acted as EDC and improved long-term allograft outcome after kidney transplantation…..]

In fact, as you suggest the dosing regimen used by Antus is rather similar to our experiment (subcutaneous administration of finasteride leads to a similar bioavailability as the higher oral dose used in my study ß I included this information in the Discussion), and additionally involved in the Discussion that the observed differences in glomerulosclerosis, tubulointerstitial fibrosis and the infiltration of mononuclear cells are because of compound flutamide/finasteride treatment.

Possibly, this mistake (about level of T) could be the result of misunderstanding what I wanted to say in this sentence. Now I changed the sentence - In our study, it can be hypothesized that the inhibition of the 5alpha-reductase leads to an decrease blood testosterone, because free T could be more willingly and quickly aromatized to estradiol, known as a strong mitotic inducer, and therefore we observed enhanced proliferative activity of the cells.

Minor comments:

I changed the immunoexpression into just expression as you suggested, to avoid misunderstanding, misinterpretation of the study

I added to Abstract the full name of used abbreviations.

I changed 10 nM into 10 mM (it was oversight, silly mistake)

The purpose of the presence of Figure 1 is to make easier understanding of the interpretation of the conducted evaluation of the TUNEL and PCNA analysis results. That’s why I decided to left it in manuscript.

I replaced alternation by alteration.

Round 2

Reviewer 3 Report

Although the authors have made some improvements to the manuscript according to my recommendations, several issues have still not yet been resolved satisfactorily:

-          The IHC figures are still not up to publishable standards. Although the differences in brightness/contrast have somewhat been improved upon since the first submission, there still is considerable difference in lighting parameters between images of the same staining, making it difficult to interpret relevant differences. This is especially the case for Fig. 2B vs. 2b, Fig. 4A vs. 4a, Fig. 7A-B vs. 7C, S1A vs. S1a, S2A vs. S2a, and S2B vs. S2b. It is very important to set the microscope lighting and camera exposure levels to an appropriate value for the first image of a set of stained slides, and to keep the same setting for the other images within the same experiment. It appears that the authors did not adhere to this elementary principle of microscopy. Many online resources exist to assist with setting up the ideal parameters to obtain the best quality microscope images; it is suggested the authors make use of these.

-          In the case of Figures 2B, S1, S2, and S3 there is a “vignetting” artefact showing more intense lighting in the center, and darker areas in the corner due to improper light path setup. Shading correction needs to be applied to these images to remove this artefact.

-          Importantly, the resolution of the histology images is below every acceptable standard. Most likely the authors used a heavily compressed JPEG format to construct the figures for the manuscript, which led to a severe loss in image quality. The resolution is so poor that individual cells can hardly be distinguished, and the scale bar text is almost illegible. This needs to be improved by supplying high-resolution images of at least 300 dpi for publication (ideally an uncompressed format such as TIFF will be used), as indicated in the instructions to authors section available on the IJERPH website.

-          Regarding the androgen receptor and junctional protein expression levels (revised tables 2 and 3), the authors did not adjust their qualitative scoring system, still relying on a subjective scale ranging from “–“ to “+++”. According to the methods, this score is the result of the evaluation of sections from 5 animals per group, but it is not clear how these results are averaged into the arbitrary “-“ to “+++” scoring system. While I understand that it is not straightforward to count positive cells for this staining, perhaps it would be better to convert the existing scoring system to numerical values. This would be very straightforward to do using the data the authors already have in hand using a simple conversion system for the staining intensity per animal: “-“ = 0; “+/-“ = 1; “+” = 2; “++” = 3; “+++” = 4. These ordinal data can then be subjected to statistical evaluation using a non-parametric test (e.g. Kruskall-Wallis).

-          The authors did not address my original question why T-tests were used to compare multiple groups, while this is not the appropriate statistical method for this type of data.

-          It is still not entirely clear how the collagen fiber thickness was measured. The data will be very dependent on the “random” selection of the region of interest within a section where the measurement is performed. Is the average width of the fiber measured? It would be better to try to measure the total PicroSirius orange or yellow/green staining area within an entire image using an unbiased image analysis tool (e.g. using the free ImageJ software).

-          I understand that a rescue experiment using DHT might fall beyond the scope of this study, but at least this approach should be highlighted in the discussion. Perhaps the authors can add a “Limitations of the study” section where they mention that this DHT rescue experiment needs to be done to confirm their hypothesis.

-          Similarly, the justification of the use of thiopental as anesthetic for hormone measurements could be mentioned in this “limitations” section.

-          In my original comments I meant that the authors more clearly need to highlight that the study from Antus et al. was performed in rats that had renal transplants, in contrast to the current study. As this is a major difference, it should also be added to the list of possible reasons to explain the discrepancies between both studies in lines 334-336.

Finally, as a general recommendation for future responses to reviewers the authors are urged to copy the full text of the reviewer’s comments into their response (e.g. in bold font), and insert their responses point-by-point within this text to make it easier for the reviewer to double-check whether all issues have been addressed properly.

Author Response

Dear Reviewer III,

I would like to thank you for you revision, very much. I was trying to improve the manuscript according to your suggestions, but some of these are out of my possibility such as performing the digital analysis of image. That’s why, we done other morphological analysis based on calculation the positive cells/structures in the histological slide, please see in the text of manuscript.

          The IHC figures are still not up to publishable standards. Although the differences in brightness/contrast have somewhat been improved upon since the first submission, there still is considerable difference in lighting parameters between images of the same staining, making it difficult to interpret relevant differences. This is especially the case for Fig. 2B vs. 2b, Fig. 4A vs. 4a, Fig. 7A-B vs. 7C, S1A vs. S1a, S2A vs. S2a, and S2B vs. S2b. It is very important to set the microscope lighting and camera exposure levels to an appropriate value for the first image of a set of stained slides, and to keep the same setting for the other images within the same experiment. It appears that the authors did not adhere to this elementary principle of microscopy. Many online resources exist to assist with setting up the ideal parameters to obtain the best quality microscope images; it is suggested the authors make use of these.

The mentioned above figures were corrected.

-          In the case of Figures 2B, S1, S2, and S3 there is a “vignetting” artefact showing more intense lighting in the center, and darker areas in the corner due to improper light path setup. Shading correction needs to be applied to these images to remove this artefact.

The vignetting artefact was removed.

-          Importantly, the resolution of the histology images is below every acceptable standard. Most likely the authors used a heavily compressed JPEG format to construct the figures for the manuscript, which led to a severe loss in image quality. The resolution is so poor that individual cells can hardly be distinguished, and the scale bar text is almost illegible. This needs to be improved by supplying high-resolution images of at least 300 dpi for publication (ideally an uncompressed format such as TIFF will be used), as indicated in the instructions to authors section available on the IJERPH website.

All figures now are shown in uncompressed format (Tiff) and the resolution of them is 600 dpi.

-          Regarding the androgen receptor and junctional protein expression levels (revised tables 2 and 3), the authors did not adjust their qualitative scoring system, still relying on a subjective scale ranging from “–“ to “+++”. According to the methods, this score is the result of the evaluation of sections from 5 animals per group, but it is not clear how these results are averaged into the arbitrary “-“ to “+++” scoring system. While I understand that it is not straightforward to count positive cells for this staining, perhaps it would be better to convert the existing scoring system to numerical values. This would be very straightforward to do using the data the authors already have in hand using a simple conversion system for the staining intensity per animal: “-“ = 0; “+/-“ = 1; “+” = 2; “++” = 3; “+++” = 4. These ordinal data can then be subjected to statistical evaluation using a non-parametric test (e.g. Kruskall-Wallis).

AR-positive cells (regardless of localization in cell and intensity of IHC reaction) were counted separately in PCT, DCT and RC and were given as a percentage from all cells number in each structure. The portion of the nephrons (PCT, DCT and RC) positive for junctional protein were counted also separately, and according the intensity of brown color of IHC reaction, the mentioned above structures have been given the score: negative (0), weak (1), moderate (2), strong (3) and very strong (4).

The receptor translocation from cytoplasm to nucleus was described within the text and showed in Fig 2.

The results confirming junctional protein expression are shown in Table 3.

-          The authors did not address my original question why T-tests were used to compare multiple groups, while this is not the appropriate statistical method for this type of data.

All Mann-Whitney U-tests and Student t-tests were used to compare the same two groups (Fin vs Control, n=5 for each), and no other groups or subgroups are compared in our paper, so the application of these simple statistical tests is justified.

That’s why the Table 5 was reorganized.

-          It is still not entirely clear how the collagen fiber thickness was measured. The data will be very dependent on the “random” selection of the region of interest within a section where the measurement is performed. Is the average width of the fiber measured? It would be better to try to measure the total PicroSirius orange or yellow/green staining area within an entire image using an unbiased image analysis tool (e.g. using the free ImageJ software).

In the experiment the thickness of collagen fibers was measure in the region of microphotographs where the fibers were visible. The microphotographs were chosen randomly, and the average of results were statistically processing.

Additionally, we done the evaluation of percentage of areas that contain fibers in relation to the entire area (as a 100%) of the slide. Morphometric valuations (thickness and area occupied by collagen) were made by Leica LAS v4.4 Core Analysis software. These results were included in to Table 5.

-          I understand that a rescue experiment using DHT might fall beyond the scope of this study, but at least this approach should be highlighted in the discussion. Perhaps the authors can add a “Limitations of the study” section where they mention that this DHT rescue experiment needs to be done to confirm their hypothesis.

In to the conclusion part was added the information about “the limitations of the study”, please find in text.

-          Similarly, the justification of the use of thiopental as anesthetic for hormone measurements could be mentioned in this “limitations” section.

In the Materials and Methods section was added the sentence: However, the thiopental is believed to modify sex hormone levels, the possible changes in these parameters were the same in both groups of animals, therefore correlation the results from one group (Control) to the other (Fin) was possible..

-          In my original comments I meant that the authors more clearly need to highlight that the study from Antus et al. was performed in rats that had renal transplants, in contrast to the current study. As this is a major difference, it should also be added to the list of possible reasons to explain the discrepancies between both studies in lines 334-336.

In the mentioned lines of Discussion was added: ”…. and animals after kidney transplantation were also given the antibiotics and immunosuppressive drugs.”  

Finally, as a general recommendation for future responses to reviewers the authors are urged to copy the full text of the reviewer’s comments into their response (e.g. in bold font), and insert their responses point-by-point within this text to make it easier for the reviewer to double-check whether all issues have been addressed properly.

I accept your recommendation with gratitude.
I tried to prepare this answer to the Reviewer with your suggestions.